# Strategies to Combat Multidrug-Resistant and Persistent Infectious Diseases

**DOI:** 10.3390/antibiotics9020065

**Published:** 2020-02-06

**Authors:** Olga Pacios, Lucia Blasco, Inès Bleriot, Laura Fernandez-Garcia, Mónica González Bardanca, Antón Ambroa, María López, German Bou, Maria Tomás

**Affiliations:** 1Microbiology Department-Research Institute Biomedical A Coruña (INIBIC), Hospital A Coruña (CHUAC), University of A Coruña (UDC), 15009 A Coruña, Spain; olgapacios776@gmail.com (O.P.); luciablasco@gmail.com (L.B.); bleriot.ines@gmail.com (I.B.); laugemis@gmail.com (L.F.-G.); monica.gonzalez.bardanca@sergas.es (M.G.B.); anton17@mundo-r.com (A.A.); maria.lopez.diaz@sergas.es (M.L.); german.bou.arevalo@sergas.es (G.B.); 2Study Group on Mechanisms of Action and Resistance to Antimicrobials (GEMARA) of Spanish Society of Infectious Diseases and Clinical Microbiology (SEIMC), 28003 Madrid, Spain; 3National Infection Service Laboratories, Public Health England, Colindale NW95EQ, UK; 4Spanish Network for the Research in Infectious Diseases (REIPI), 41071 Sevilla, Spain

**Keywords:** anti-MDR strategies, anti-persistent treatments, drug repurposing

## Abstract

Antibiotic failure is one of the most worrying health problems worldwide. We are currently facing an international crisis with several problematic facets: new antibiotics are no longer being discovered, resistance mechanisms are occurring in almost all clinical isolates of bacteria, and recurrent infections caused by persistent bacteria are hampering the successful treatment of infections. In this context, new anti-infectious strategies against multidrug-resistant (MDR) and persistent bacteria, as well as the rescue of Food and Drug Administration (FDA)-approved compounds (drug repurposing), are being explored. Among the highlighted new anti-infectious strategies, in this review, we focus on antimicrobial peptides, anti-virulence compounds, phage therapy, and new molecules. As drugs that are being repurposed, we highlight anti-inflammatory compounds, anti-psychotics, anti-helminthics, anti-cancerous drugs, and statins.

## 1. Introduction

Antimicrobial resistance is currently considered one of the principal threats to global public health by the World Health Organization (WHO) [1], especially because of the global spread of multidrug-resistant (MDR) bacterial pathogens. MDR pathogens can develop resistance to different antimicrobials via horizontal gene transfer and via gene mutations as a consequence of exposure to these drugs. Although the acquisition of resistance is a natural process, it is exacerbated by the misuse of antibiotics, inadequate surveillance, and the poorly controlled regulation of antibiotics in clinical medicine and in the livestock industry, which led to the appearance and spread of MDR bacteria all over the world [1,2,3].

Mortality caused by resistance to antibiotics is a major health problem, causing more than 23,000 deaths a year in the United States (US) alone (Center for Disease Control, Antibiotic Resistance Threats in the United States, 2019 (2019 Antibiotic Resistance Threats Report)) and more than 33,000 in Europe [4]. Already alarming, the number of deaths is predicted to increase worldwide to more than 10 million by 2050, although predictions are difficult to make in this field [5,6].

In 2008, Rice and colleagues first referred to the six most frequent MDR bacterial pathogens, i.e., *Enterococcus faecium*, *Staphylococcus aureus*, *Klebsiella pneumoniae*, *Acinetobacter baumannii*, *Pseudomonas aeruginosa*, and *Enterobacter* spp., as the “ESKAPE” pathogens [7]. Some other opportunistic pathogens, such as *Escherichia coli*, *Enterococcus faecalis*, and *Burkholderia cepacia* can also become MDR strains and are able to cause severe infections.

Bacteria can evade the antimicrobial activity of antibiotics via three different but related mechanisms: resistance, tolerance, and persistence [8]. Resistance is the ability of bacteria to grow in the presence of high concentrations of an antibiotic and is caused by inherited mutations, which affect the efflux pumps, the drug target, or the antibiotic molecule itself [8,9]. Resistant populations can be found in all types of environments: water, animals, inanimate surfaces, humans, plants, and food [10,11,12,13]. Moreover, resistant bacteria can grow under antibiotic pressure, their resistant phenotype is inheritable, and a significant increase in the minimal inhibitory concentration (MIC) of antibiotics is required to effectively kill them [14].

However, resistance phenotypes are not always due to the acquisition of resistance genes or mutations in the bacterial genome, and they can often explained by the appearance of a persistent phenotype, which includes tolerant populations of bacteria and/or the presence of persistent sub-populations (known as “persister cells” or “persisters”) within a population of susceptible cells (i.e., killed by an antibiotic at a concentration equal to or lower than the MIC). Bacteria exhibiting this persistent phenotype are able to overcome antibiotic treatment but do not affect the MICs of the drugs. As with resistance, tolerance and persistence were first observed shortly after the introduction of penicillin, as reviewed very recently by Windels and colleagues [15]. Antibiotics then become ineffective owing to the lack of cellular metabolism and to the effect on or interference in DNA replication, transcription, and translation, as well as cell-wall synthesis. “Persisters” are non-growing, metabolically inactive, dormant bacteria that exhibit transient high levels of tolerance to antibiotics and play a non-negligible role in chronic or recurrent infections [14], as they can survive both antibiotic therapy and host immune responses. The dormant bacteria can rapidly revert to the wild-type (WT) phenotype and regain their antibiotic susceptibility when drug pressure is removed and their metabolic activity is reactivated. The signaling pathways involved in this “awaking” process are being further investigated [16,17]. Persister cells can survive in immune-compromised patients but also in patients in whom antibiotics do not effectively kill pathogenic bacteria via immune-evasion strategies [16]. There is currently strong evidence suggesting that the ability of bacteria to live inside some cells (such as macrophages) and the formation of biofilms are both associated with persistent infections [18]. Furthermore, clinical isolates from chronic infections caused by *Candida albicans* [19], *P. aeruginosa* [20], or uropathogenic *E. coli* [21] exposed to antibiotic pressure during a long period of time are also associated with persistent infections, exhibiting increased persister levels relative to isolates from acute or early-stage infections.

The presence of persisters in common bacterial infections is reported in patients and linked to relapses of infection. Mulcahy and colleagues isolated clinical strains of *P. aeruginosa* from the lungs of cystic fibrosis patients and observed increasing persister levels during antibiotic treatment [20]. Schumacher and colleagues showed that high-persister *hipA* mutations in *E. coli* were selected in recurrent urinary infections over time and also observed the importance of the *hipA7* mutation in persister formation in vitro [22].

Tolerant bacteria also exhibit an antibiotic-resistant phenotype. Tolerance is defined as the ability to survive transient exposure to high concentrations of antibiotics, and it can be inherited or not [9]. Unlike persistent subpopulations, tolerant bacteria are metabolically active, although their vital processes were shown to slow down [23].

Both tolerant and persistent cells are currently underestimated by the scientific community, despite evidence showing that they promote the evolution of resistance in bacteria. As early as 1988, Moreillon and Tomasz demonstrated that cyclic exposure of pneumococcus (*Streptococcus pneumoniae)* to high concentrations of penicillin selects for tolerant mutants, while resistant mutants evolve during exposure to low (but sustained) concentrations of penicillin [24]. One decade later, Novak and colleagues established that clinical isolates of vancomycin-tolerant pneumococcus had mutations in the *vncS* gene and that streptomycin- or penicillin-tolerant *S. pneumoniae* mutant showed greater efficiency in transformation with DNA compared to a WT *S. pneumoniae* strain [25]. This finding consolidated the idea of tolerant bacteria facilitating the acquisition of resistance genes via horizontal transfer, e.g., via transformation.

In 2017, two important studies shed some light on the mechanisms driving the evolution of resistance by persisters and tolerant bacteria. Firstly, in studies involving long exposure of *Mycobacterium tuberculosis* to rifampin, Sebastian and colleagues concluded that persisters are a source of de novo resistant mutants [26]; secondly, Balaban and colleagues observed that the evolution of increased tolerance or persistence confer mutations in *E. coli* populations under intermittent ampicillin exposure [27]. In 2019, Windels and colleagues discovered a strong, positive correlation between persister levels and the probability of resistance evolving in natural isolates and laboratory strains of *E. coli* [28]. Although different types of antibiotics and different experimental conditions were used in these three studies, the results are consistent, strongly suggesting a widespread link between persistence/tolerance to antibiotics and the evolution of resistance to these drugs. Considerable efforts should hereafter be devoted to the development of strategies that can eliminate tolerant and persistent cells, which potentially favor the evolution of resistance [28].

Given the immense number of possible strategies for treating MDR and persistent bacteria, it is time to examine what was achieved so far by the different therapies available. Thus, the present review aims to gather the results of the most relevant studies, organized by type of drug administered or therapeutic strategy, in order to aid researchers in the fight against infectious diseases caused by MDR and persistent bacteria.

Throughout this review, the most commonly studied strategies used against MDR bacteria and/or persisters are presented in two sections. In the first section, we present new therapies, including antimicrobial peptides (AMPs), anti-virulence compounds such as quorum sensing (QS) inhibitors, phages, and some new, chemically synthesized molecules, such as vitamin-A derivatives (retinoids). In the second section, we summarize the most important drug repurposing strategies (use of anti-inflammatories, anti-psychotics, anti-parasitics, anti-cancerous drugs, and statins) with potential antibacterial effects. These approaches are summarized in Figure 1.

Interest in these therapies or strategies is increasing, as illustrated by the two new research topics in which we recently participated, which focus on the quorum network of MDR pathogens [29] and drug repurposing for bacterial and viral infections [30]. To shed further light on trends of concern to scientists around the globe, Figure 2 summarizes the number of annual publications available on the PubMed search engine per decade, from the 1940s until the present day.

## 2. New Anti-Infectious Treatments against MDR and Persistent Bacteria

In this section, we highlight various new anti-infectious treatments against MDR and persistent bacteria.

### 2.1. Antimicrobial Peptides (AMPs)

Antimicrobial peptides (AMPs) are currently being studied and developed for the treatment of recalcitrant bacterial infections, which are often caused by persistent bacteria. In relation to this relatively new therapy, the synergistic interactions between AMPs and antibiotics together with the anti-biofilm activity of AMPs are of great interest. Many AMPs are produced naturally by a broad range of organisms, while others are designed and chemically synthesized in the laboratory.

In a recent study, Yang and colleagues characterized nine fungal defensin-like peptides, i.e., peptides synthesized by neutrophils with antimicrobial and cytotoxic properties, and identified them as potent antibacterial compounds [31]. The peptides were expressed in *Pichia pastoris*, and their antibacterial and anti-biofilm activities against MDR and persistent *S. aureus* were evaluated. The study findings showed that a defensin-like peptide called P2 exhibited the highest activity and expression level with low toxicity, no resistance, high stability, and a MIC of <2 μg/mL. P2 bound to bacterial DNA, wrinkled the outer membrane, permeabilized the cytoplasmic membrane, and inhibited *S. aureus* biofilm formation. Importantly, P2 killed 99% of persistent bacteria, which were resistant to 100× MIC of vancomycin. These data suggest that P2 may be a candidate in the development of novel antimicrobial agents against MDR and persistent staphylococcal infections.

Li and colleagues characterized two novel peptides, P5 (YIRKIRRFFKKLKKILKK-NH2) and P9 (SYERKINRHFKTLKKNLKKK-NH2), which exhibit potent antimicrobial activities against both methicillin-sensitive *S. aureus* clinical isolates and methicillin-resistant *S. aureus* (MRSA) strains [32]. As previously described for P2, these peptides did not cause significant hemolysis or cytotoxicity to renal epithelial cells. P5 and P9 significantly inhibited the *S. aureus* biofilm and disrupted the cell membrane, in addition to downregulating several virulence genes. P5 and P9 may, therefore, be promising antibacterial agents for the treatment of MRSA infections.

In 2019, Liu and colleagues investigated the potential applications of cationic peptides in the fight against vancomycin-resistant *Enterococcus* (VRE) [33] and found some peptides displaying moderate bactericidal activity against VRE, with MIC values of 2–8 μg/mL, as well as significant synergistic interactions between these peptides and vancomycin. The mechanism of action of these peptides is the inhibition of *vanRS* transcription, a key two-component system (TCS) in the resistance against vancomycin. These researchers showed that inhibition of *vanRS* transcription restored vancomycin activity. Consistent with in vitro results, the researchers observed higher survival rates in *Galleria mellonella* larvae treated with these cationic peptides plus vancomycin than in larvae treated with vancomycin alone. Together, these results suggest that cationic peptides show antibacterial activity against VRE and also reverse vancomycin resistance in *Enterococcus*, and they are, therefore, promising candidates for combating vancomycin-resistant pathogens.

### 2.2. Anti-Virulence Compounds

Anti-virulence compounds inhibit bacterial virulence factors, thus preventing the development of infection without affecting bacterial growth. In 2013, Pan and colleagues characterized a chemical compound called BF8 ((*Z*)-4-bromo-5-(bromomethylene)-3-methylfuran-2(5*H*)-one) that was able to reduce persistence of *E. coli* growth and revert the antibiotic tolerance of persisters [34]. BF8 is a QS inhibitor of *E. coli* which disrupts *E. coli* biofilms and renders associated cells more sensitive to ofloxacin. BF8 appeared safe to mammalian cells in vitro and showed no long-term cytotoxicity in vivo in a healthy mouse model. 

In 2013, Conlon and colleagues developed an acyldepsipeptide antibiotic, called ADEP4, which can activate the ClpP protease and, thus, cause the death of persistent bacterial cells by degrading around 400 proteins [35]. In addition, this ADEP4 was also found to be active against persisters in both planktonic and biofilm states. A combination of ADEP4 with rifampicin completely killed *S. aureus* biofilms in vitro and in a chronic infection murine model in vivo.

The first compounds able to reduce the formation of antibiotic-tolerant persister cells of *P. aeruginosa* were identified by Starkey and colleagues [36]. Some of these compounds, such as M64, blocked the synthesis of both pro-persistence and pro-acute MvfR-dependent signaling molecules. MvfR, also referred to as PqsR, is the global virulence QS transcriptional regulator of *P. aeruginosa*. M64 was found to be active against MDR strains of *P. aeruginosa* that cause acute and persistent murine infections and it did not perturb bacterial growth, which limited selective resistance.

Regarding *E. coli,* El Meouche and colleagues [37] demonstrated that the AcrAB–TolC apparatus in *E. coli* increases the DNA mismatch repair system, promoting spontaneous mutations that can lead to high-level resistance to antibiotics. Thus, inhibitors of the AcrAB–TolC efflux pump could not only re-activate the antibiotic sensitivity of *E. coli* cells but could also restore their mutation rates, potentially improving current treatments.

### 2.3. Phage Therapy Alone or in Combination with Antibiotics to Treat MDR and Persistent Bacteria

Clearly some alternative antibacterial therapies must emerge in order to halt the spread of MDR organisms. In this context, phage therapy is an antibacterial approach that involves using natural bacterial viruses (known as bacteriophages or phages) that infect and lyse bacteria to treat or prevent infectious disease [38,39]. These viruses are natural predators of bacteria and, as recently reviewed by Divya and colleagues [40], phages were used as a primary treatment for bacterial diseases from their discovery in the 1900s for a period of 25 years, until they were absolutely eclipsed by antibiotics.

One of the advantages of phage therapy over broad-spectrum antibiotics is the high specificity toward target bacterial pathogens, with no adverse effects on the host itself or the host commensal microbiota [41], thus minimizing secondary effects. One interesting aspect of phage therapy is the synergistic interactions between phages and the host immune system [42]. Indeed, as it is not beneficial for phages to kill all of the host bacteria at the infection site (otherwise, they could not continue to replicate), they can be expected to control the bacterial pathogens and significantly reduce the population, thus providing the patient’s immune system the chance to eliminate the remaining pathogens. Here, we cite a few examples of phage-based treatments of MDR and persistent infections.

In 2017, Schooley and colleagues reported that a 68-year-old diabetic patient with necrotizing pancreatitis complicated by an MDR *A. baumannii* infection was successfully treated with a personalized phage-based therapy consisting of nine lytic phages [43]. Five days after the bacteriophage-based therapy was started, once the infection was controlled, minocycline was administered, and infection was finally overcome.

In 2011, Khawaldeh et al. described the successful use of six lytic *P. aeruginosa* phages as an adjunctive therapy for a recurrent bladder infection in a 67-year-old woman, most likely caused by a persistent subpopulation of *P. aeruginosa* cells [44]. Antibiotics alone failed to overcome the infection, as is usually the case with recalcitrant, persistent bacteria. However, the combined use of meropenem and colistin and this phage cocktail led to the improvement of symptoms and a significant reduction in bacterial load, and the treatment was well-tolerated by the patient. The cocktail was applied every 12 h for 10 days and, from day eight onward, the bacterial count decreased until no viable bacteria were longer detected.

Studies carried out by research teams led by Torres-Barceló [45] and Chaudhry [46] showed that the therapeutic effect was improved by starting treatment with antibiotic once the phages already started to fight infections caused by *P. aeruginosa*. Both of these studies highlight the importance of the sequential application of both treatments in order to achieve good results, as it appears that bacterial pathogens become more vulnerable to some antibiotics after phage treatment [47] and that the phages can even restore their sensitivity to antibiotics [48]. Importantly, phages can use bacterial efflux pumps as receptors, which would affect the clearance of antibiotics, one of the most widespread resistance mechanisms of bacteria [47].

In 2019, Blasco and colleagues described an engineered lysogenic phage (Ab105-2phiΔCI) in which repressor CI protein was deleted, thus becoming lytic [49]. These researchers used a sophisticated assay in the first study using a mutant lytic phage in combination with imipenem and meropenem antibiotics. The combination caused a significant decrease in *A. baumannii* MDR cells both in vitro and in vivo, with lower MICs needed for these carbapenems to be effective against the resistant pathogen.

The anti-biofilm activity of phage phi11 endolysin, which can kill staphylococcal biofilms via its two endopeptidase domains, was described for the first time in 2007 [50]. Research teams led by Fenton and Shen described lytic endolysins (CHAP(K) and PlyC) which successfully removed staphylococcal and streptococcal biofilms, respectively [51,52]. As biofilms are mainly composed of persistent cells, the use of phage-derived endolysins may represent an effective “anti-persister” strategy. The lytic activity displayed by endolysin PlyE146 against *E. coli*, *P. aeruginosa*, and *A. baumannii* [53] is important as these three bacteria are important nosocomial pathogens that represent a major risk to health. Finally, LysAB2 endolysin, described for the first time in 2011, showed activity against different bacterial species, such as MRSA, *A. baumannii*, and *E. coli* [54]. Interestingly, peptide-induced modification of endolysin LysAB2 broadened the range of lytic activity [54].

### 2.4. New Molecules

In 2018, Kim and colleagues described the antibacterial activity of two synthetic retinoids (vitamin A analogues), named CD437 and CD1530 [55]. These compounds exhibited high in vivo killing rates of both growing and persister *S. aureus* cells, by disrupting lipid bilayers. The mechanism of action is as follows: the carboxylic acid and the phenolic groups anchor the retinoids to the surface of the bacterial membrane bilayer, by strongly binding to hydrophilic lipid heads. As a result, the retinoids penetrate the bilayers and become embedded orthogonally in the lipid molecules in the outer membrane leaflet, inducing substantial perturbations and permeabilizing the membranes of MRSA and persisters. In addition, both compounds displayed synergistic interactions with gentamicin, a low probability of resistance selection, and potent activity against a panel of clinical *S. aureus* and *E. faecium* strains. The major obstacle for developing retinoids as therapeutic agents is their potential cytotoxicity, which is a matter of considerable debate. However, none of the tested molecules exhibited cytotoxicity in vitro [55].

This discovery suggests the possibility of chemically changing these molecules to minimize their cytotoxicity and synthesizing more antimicrobial retinoid-based compounds. Indeed, approximately 4000 retinoid analogues were synthesized so far [56]. Deciphering the chemical properties and interactions between bacterial compounds and the candidate molecules will be essential in order to produce good antibacterial agents.

## 3. Repurposing Treatments against MDR and Persistent Bacteria

In this era of MDR pathogens, during which the discovery and development of new antibiotics are limited and generally unsuccessful, new strategies must be employed to enhance the fight against infectious diseases. Thus, the use of non-antibiotic compounds, referred to as drug repurposing and “repositioning”, is of great interest. Focusing on drug repurposing strategies that were tested with MDR bacteria shows that resistance is rarely crossed, usually because the active molecule affects a different target from the antibiotic target. Moreover, time- and economic-related advantages of FDA-approved drugs must also be considered.

In this section, we describe various repurposing treatments against MDR and persistent bacteria.

### 3.1. Anti-Inflammatories as Antibacterial Agents

Vijayashree and colleagues recently used in silico tools to study the antibacterial effects of acetaminophen and ibuprofen, anti-inflammatory, anti-pyretic, and analgesic drugs, against red-complex pathogens *(Porphyromonas gingivalis, Treponema denticola*, and *Tannerella forsythia*), as these bacteria are associated with inflammatory conditions associated with periodontal disease [57]. Acetaminophen and ibuprofen were found to interact by in silico approximations with bacterial cytoplasmic proteins involved in cellular processes, metabolism, and virulence. The authors claimed that bioinformatic prediction tools revealed multiple epitopes in the virulent proteins that should be targeted in in vitro assays.

Glucocorticoids are a well-known class of anti-inflammatory drugs, but their use in patients with sepsis is highly controversial because of the immunosuppressive effects of these compounds [58]. Betamethasone is an anti-inflammatory steroid belonging to this class. Indeed, once in the nucleus, betamethasone stimulates the transcription and translation of lipocortin and vasocortin, two proteins that inhibit the release of inflammatory mediators such as prostaglandins, leukotrienes, and histamine [59]. Although it may seem counterproductive to use these drugs in sepsis patients, as they mitigate the immune response, Emgard and colleagues showed that the topical steroid betamethasone was effective, by itself, for treating external otitis caused by *P. aeruginosa* and *C. albicans* [60]. These findings could be explained considering that inflammation is a major mechanism in the development of external otitis, similarly to what occurs in periodontal disease, as described above.

Finally, we include here a cathelicidin-like antimicrobial peptide, Cbf-K_16_, characterized by Jiang and colleagues and proposed as an anti-inflammatory and antibacterial compound that can effectively kill clarithromycin- and amoxicillin-resistant *Helicobacter pylori* SS1, both in vitro and in a gastritis mouse model [61]. Cbf-K_16_ showed time-dependent killing kinetics, protection of *H. pylori*-infected gastric epithelial cells, and inhibition of interleukin 8 (IL-8) secretion. Indeed, Cbf-K_16_ binds to genomic DNA, downregulating the expression of adhesion genes (*alpA* and *alpB*) and the virulence gene (*cagA*), thus indicating its potential use of the in the development of an anti-infective therapy.

### 3.2. Anti-Psychotics

A decade ago, Lieberman and Higgins screened the antibacterial activity of some compounds that affected neurological function, identifying 68 that disrupted infection of macrophages by *Listeria monocytogenes* [62]. After further examination of the compounds, these researchers indicated that thioridazine, an antipsychotic drug used to treat schizophrenia during the last 40 years, and bepridil, a calcium channel blocker, decreased intracellular infection by *L. monocytogenes* in a dose-dependent manner by significantly inhibiting escape from vacuoles in vitro. Treatment of host cells with thioridazine or bepridil significantly decreased the ability of *L. monocytogenes* to escape the phagocytic vacuole, a step required to initiate intracellular replication. Although experiments were performed using the WT *L. monocytogenes* strain 10403S, this is an elegant example of a drug repurposing strategy, as calcium channel blockers can be used in the treatment of brain injury, as well as to combat bacterial pathogens. The effects of bepridil and thioridazine on persistent and resistant bacteria should be evaluated. In addition, the antimicrobial activity of thioridazine against other bacterial species, such as *S. aureus* [63] and *M. tuberculosis*, was also reported [64]. In the first case, the researchers observed that a dose of only 0.1 mg/L thioridazine completely inhibited the growth of *S. aureus* phagocytosed by macrophages, suggesting intracellular bactericidal activity. Nevertheless, thioridazine was withdrawn from use worldwide in 2005 because it caused severe cardiac arrhythmia in some patients (it binds to histamine receptors). However, some generic forms of this drug are still available in the US.

In 2016, Andersson and colleagues described the antibacterial effects of another antipsychotic drug, trifluoperazine [65]. Although trifluoperazine MIC values were too high to be reached in plasma and the drug did not affect the growth of *Yersinia pestis* cells or the expression/production of their type 3 secretion system (T3SS, an important virulent factor for this bacteria), use of this antipsychotic increased the survival of *Y. pestis*-infected macrophages in vitro and the survival of infected mice in vivo. Trifluoperazine was then tested in both *Salmonella enterica* serovar Typhimurium and *Clostridium difficile* infection murine models, where once again it significantly increased the survival of infected mice. Interestingly, although the antibacterial activity is yet to be described, a significantly higher survival rate of mice infected with *Y. pestis, S. enterica*, and *C. difficile* and treated with the drug was reported. It is, therefore, possible that the bactericidal activity is intracellular, as with thioridazine, another antipsychotic drug with a similar structure.

### 3.3. Anti-Helminthic Drugs

Salicylanilides are antiseptics that are used worldwide, and they are widely studied. These compounds are thought to act by uncoupling oxidative phosphorylation, thereby impairing the motility of parasites. Rajamuthiah and colleagues described the efficacy of niclosamide (a bacteriostatic agent included in the salicylanilide family) against methicillin-, vancomycin-, linezolid-, and daptomycin-resistant *S. aureus* isolates, and they concluded that they probably damage the bacterial membrane [66]. Niclosamide inhibits QS and virulence genes in *P. aeruginosa* [67], such as phospholipase C, LasA protease, pyocyanin, chitinase, and rhamnolipids. It also increases the negative charges on the cell walls of *A. baumannii* and *K. pneumoniae*, resulting in synergistic interaction with cationic colistin, resensitizing both pathogens to this antibiotic [68]. These studies show consistent results although they were performed with four different bacterial species and conditions.

In 2016, Gooyit and Janda reported that other members of the salicylanilide family, such as rafoxanide and closantel, presented greater bactericidal activity against the logarithmic and stationary phases of *C. difficile* than vancomycin [69]. Avermectins, a group of broad-spectrum anti-helminthics, demonstrated in vitro efficacy against *M. tuberculosis* and *Mycobacterium ulcerans*, with MIC values ranging from 1 to 8 mg/L and 4 to 8 mg/L, respectively [70]. Ashraf and colleagues observed the bacteriostatic effect of ivermectin, an avermectin, against clinical isolates of *S. aureus* in vitro [71]. Ten years previously, Zhang and colleagues reported that ivermectin improved the survival of mice challenged by lethal doses of lipopolysaccharide (LPS), significantly reducing the levels of tumor necrosis factor α (TNF-α), IL-1b, and IL-6 [72]. These researchers observed the same findings in vitro. Ivermectin also blocked the nuclear factor kappa-light-chain-enhancer of activated B-cells (NF-κB) pathway and reduced endotoxemia (presence of endotoxins in the blood) and the associated inflammation.

### 3.4. Anti-Cancerous Drugs as Antibacterials

The use of anti-cancer agents to treat bacterial pathogens may appear surprising. However, although cancer and infectious diseases differ in many aspects, drug-tolerant persisters also occur in cancer cell populations and are implicated in the recurrence of tumors.

Cheng and colleagues recently used an *A. baumannii* strain resistant to most antibiotics (AB5075) and reported that three antineoplastics (5-fluorouracil, 6-thioguanine, and pifithrin-μ), an anti-rheumatic (auranofin), an antipsychotic (fluspirilene), an anti-inflammatory (Bay 11-7082) and an alcohol detergent (disulfiram) inhibited the growth of MDR *A. baumannii* [73]. The best candidates among all the repurposed drugs in the treatment of MDR clinical *A. baumannii* appeared to be 5-fluorouracil and 6-thioguanine, as the inhibitory concentration 90 (IC_90_) values and MIC were lower than standard plasma drug concentration levels in human, suggesting possible use without major adverse events. The authors hypothesized that 5-fluorouracil may display the same inhibitory mechanism against bacterial pathogens as against tumor cells, by inhibiting thymidylate synthase. Similarly, 6-thioguanine may also share its mechanism of action against tumor cells and bacteria, as it works as a guanine analogue, disrupting DNA and RNA synthesis.

The antibacterial activity of the metal gallium is known for more than 80 years. Due to its chemical similarity to iron, gallium inhibits ferric redox reactions or pathways affecting bacterial growth. Gallium has a broad spectrum of activity, particularly against MDR ESKAPE pathogens [74,75]. In fact, the activity of gallium was assessed in a phase 2 trial in cystic fibrosis *P. aeruginosa*-infected patients [76], providing evidence of its safety and efficacy for human infections, improving pulmonary capacity without affecting the activity of essential human enzymes, such as superoxide dismutase and aconitase.

Mitomycin C is also known to induce DNA cross-linking in a growth-independent manner, killing persisters and actively growing cells of several pathogens such as *E. coli*, *S. aureus*, and *P. aeruginosa* [77]. However, Chowdhury and colleagues claimed that the anti-cancer drug cisplatin forms intra-strand DNA cross-links and, therefore, eradicates *E. coli* K-12, *S. aureus*, and *P. aeruginosa* persister cells through a growth-independent mechanism and more efficiently than mitomycin C, which forms inter-strand DNA cross-links [78].

Finally, hormonal modulators used as anti-cancerous agents also play a role in the fight against resistant bacterial populations. Selective estrogen receptor modulators (SERMs) are widely used to treat breast cancer. These drugs include clomiphene, currently under preclinical development for fertility treatment [79]. Clomiphene displays in vitro activity against *S. aureus*, with an MIC value of 8 mg/L. It acts by inhibiting undecaprenyl diphosphate synthase (UPPS), an enzyme involved in the synthesis of teichoic and peptidoglycan of the *S. aureus* cell wall. It was also reported that, due to its mode of action, clomiphene interacts synergistically with β-lactams in restoring MRSA susceptibility [79].

### 3.5. Statins

Statins are lipid-lowering molecules with pleiotropic effects, and their potential antibacterial ability was reported. In 2008, Jerwood and Cohen observed that some statins directly inhibited the growth of species belonging to the genera *Staphylococcus*, *Streptococcus*, *Enterococcus*, and *Moraxella*, thus indicating some possible antimicrobial activity of the drugs [80]. In humans, statins inhibit the enzyme 3-hydroxy-3-methyl-glutaryl-coenzyme A reductase (HMG-CoA) leading to decreased synthesis of cholesterol and increased removal of low-density lipoprotein (LDL) circulating in the body [81].

Graziano and colleagues evaluated the antibacterial activity of simvastatin, and they observed no synergistic interactions between the statin and vancomycin [82]. However, simvastatin was able to reduce the formation and viability of mature biofilms, decreasing cell viability and extra-polysaccharide production. Simvastatin also caused a significant decrease in *M. tuberculosis* bacterial load, presumably by reducing cholesterol synthesis due to the inhibition of HMG-CoA reductase within the phagosomal membrane (reviewed in Reference [83]).

Using a murine MRSA skin infection model, Thangamani and colleagues confirmed that simvastatin significantly reduced the bacterial burden in infected wounds, displaying excellent anti-biofilm activity against established staphylococcal biofilms in vivo [84]. Considered together, these findings suggest the potential bactericidal activity of simvastatin alone or in combination with topical antimicrobials currently used to treat MRSA skin infections.

The in vitro antibacterial effects of statins against skin pathogens such as *S. aureus*, *E. coli*, *P. aeruginosa*, and *Serratia marcescens* were verified in a recent study [85].

Table 1 and Table 2 respectively summarize the anti-MDR bacteria and anti-persister treatments described above.

## 4. Discussion

Bacterial infections account for millions of deaths annually worldwide and, among these, persistent infections have a large clinical impact. Persistent cells are one of the main causes of such recurring infections [86]. As stated above, persistence to antibiotics is not always due to genetic changes and can be caused by metabolic inactivity and dormancy, strictly regulated by complex molecular processes such as the (p)ppGpp network, QS, or toxin–antitoxin (TA) systems, all of which were recently reviewed by Trastoy and colleagues [23]. Strategies for avoiding antibiotic therapy failure usually focus on genetic resistance, while other bacterial survival strategies, such as persistence or tolerance to antibiotics, are increasing. A closer look at such strategies is necessary in order to combat persistent and re-incident bacterial infections caused by resistant pathogens. In an interesting recent review, Theuretzbacher and colleagues examined some of the new treatments summarized in the present review, together with vaccines and immune-modulator-based therapies [87].

In the present review, we focused first on new treatments (antimicrobial peptides, anti-virulence strategies, phage-based therapies, and new molecules) and then on drug repurposing assays (anti-inflammatory and anti-psychotic compounds, anti-helminthics, anti-cancerous drugs, and statins). Antimicrobial peptides open a huge spectrum of possibilities as they are less prone to generating resistance than antibiotics and they can be chemically modified to enhance their antibacterial activity. We mentioned two of these (Table 1), but the mechanisms of action of several AMPs currently being investigated were reviewed by Kang and colleagues, which basically involve interference of the membrane or disruption of cellular processes [88]. Sierra and colleagues also conducted an exhaustive review of the advances regarding AMPs [89] and reported that 20 AMPs are currently being tested in clinical trials, ranging from preclinical stage to phase III, mostly, although not exclusively, for topical indications. The vast majority of these compounds are cyclic polycationic peptides.

Focusing on anti-virulence strategies, we included here QS inhibitors, inhibitors of global virulence regulators, and pump-efflux inhibitors, as well as a ClpP protease activator. Much information about these anti-virulent compounds was provided in an exhaustive review conducted by Dickey and colleagues [90]. Similarly, López and colleagues conducted a detailed analysis of 26 patents of anti-virulence compounds published between 1994 and 2012 [91]. These compounds included some inhibitors of adhesion and colonization, secretion systems, or cellular signaling systems, among other virulence factors. The large number of patents produced in less than two decades reflects the increasing interest in anti-virulence compounds and their antibacterial properties.

Regarding bacteriophages or their components (e.g., endolysins, reviewed in the text), in Western countries, there are regulatory hurdles and legal problems associated with the use and administration of viruses as clinical tools. The factors leading to the controversy surrounding the clinical use of using phages as antimicrobial agents include the complicated regulatory issues, safety concerns, and skepticism about their therapeutic efficacy, e.g., because of the evolution of phage resistance. To date, only nine trials using phage therapy were conducted [92]. As suggested by Harper in 2018, the choice of appropriate targets is an important factor to consider as the high species specificity of phages may be desirable in monomicrobial infections but can limit polymicrobial infections [93]. Phage–antibiotic synergy (PAS) was demonstrated both in vitro and in vivo [94] and, even when no benefits were obtained, the emergence of antibiotic- or phage-resistant phenotypes was greatly minimized. The combination of phage therapy and antibiotics would also be of benefit because of the improved bacterial clearance together with the reduced bacterial capacity of developing resistance to one or both therapies, as concluded by Torres-Barceló and Hochberg [95] and Tagliaferri and colleagues [96]. Moreover, as with synthetic molecules with potential antibacterial effects, phages or phage-derived endolysins can be biochemically modified to extend the range of susceptible organisms. The potential activity of bacteriophage endolysins to supplement or replace antibiotics is an exciting topic which was reviewed by Love and colleagues [97] and by Gondil and colleagues [98], who also mentioned the safety of these endolysins in humans.

Given the time involved and economic problems associated with the development of a new drugs, “rescuing” drugs already approved by the FDA that exhibit antibacterial activity is another possible option for fighting persistent infections. Non-antibiotic compounds can be effective when used in combination with other drugs or antibiotics, although further studies must be conducted to determine effective concentrations that are clinically tolerated and safe. In terms of economic issues, we agree with Leyclit and colleagues who, in a recent review article, mentioned that pharmaceutical industries have little interest in re-profiling existing drugs due to the lack of profit [99]. However, drug repurposing can have real economic advantages, as structural and pharmacological studies were already conducted, e.g., studies concerning the bioavailability or safety profiles [83]. Because toxicity and pharmacokinetics are already known, preclinical trials could be bypassed and clinical phase 2 undertaken to test the effectiveness of the drugs [100], also representing an advantage in terms of time.

Also in relation to the drug repurposing strategy, in a recent review, Gupta and colleagues discussed some of the properties of farnesol, a QS molecule described in *C. albicans* that can be used as an anti-inflammatory but also as an anti-biofilm, anti-cancer, and anti-tumor agent [101]. Similarly, Liu and coworkers reviewed the topic of triazines, nitrogen-containing heterocyclic molecules that display a wide range of pharmacological activities, such as anti-bacterial, anti-malarial, anti-human immunodeficiency virus (HIV), anti-cancer, and anti-oxidant activity [102]. The anti-bacterial activity of these compounds was tested both in vitro and in vivo.

Among the antipsychotic drugs that display anti-infective activity, thioridazine and bepridil are the most promising compounds. Some antipsychotics can act as calcium channel inhibitors. As bepridil is also a calcium channel blocker, it is clear that calcium fluxes within host cells following infection by *L. monocytogenes* are involved in the entry of bacteria into and escape from vacuoles. As observed with trifluoperazine, which is effective against *Y. pestis, S.* Typhimurium, and *C. difficile*, these drugs may have a broad spectrum as many pathogens rely on similar mechanisms to modulate virulence or host pathways [65].

Regarding anti-helminthics, niclosamide seems to show the best results in vitro, and it is efficient against a broad spectrum of bacterial pathogens, such as *P. aeruginosa*, *S. aureus*, *K. pneumoniae*, *A. baumannii*, and *H. pylori*. Niclosamide also showed therapeutic efficacy in an experimental infection model of *Galleria mellonella* larvae infected with *P. aeruginosa* and *H. pylori* [67,103], although further in vivo experiments in vertebrates must be conducted to determine the dose that can be used without causing major adverse effects. A formulation of niclosamide under nanosuspension showed lower toxicity in a rat lung infection model involving *P. aeruginosa*, and it was recommended that this formulation should undergo further study [104].

Finally, some statins were also tested as possible anti-bacterial treatments, with good results due to their capacity to attenuate virulence factors, interfering with teichoic and lipoteichoic acids and disrupting cellular structures. In a recent review article, Ko and colleagues [105] compared the antibacterial activity of several statins and concluded that simvastatin appears the most promising for use as an antibiotic adjuvant.

Thus, repurposing approved drugs may be highly effective against multiple antibiotic-resistant pathogens, taking into account the current (increasing) problem of antimicrobial ineffectiveness and resistance [83,99].

## 5. Conclusions

In summary, we believe that one possible line of study in the fight against persistent infectious bacteria resides in analyzing how several networks associated with molecular mechanisms of bacterial tolerance or persistence are coordinated. The combination of new anti-infectious treatments, as well as drug repurposing alone or in association with antimicrobials, may represent an efficient way of combating multidrug resistant (MDR) and persistent infectious bacteria.

## Figures and Tables

**Figure 1 antibiotics-09-00065-f001:**
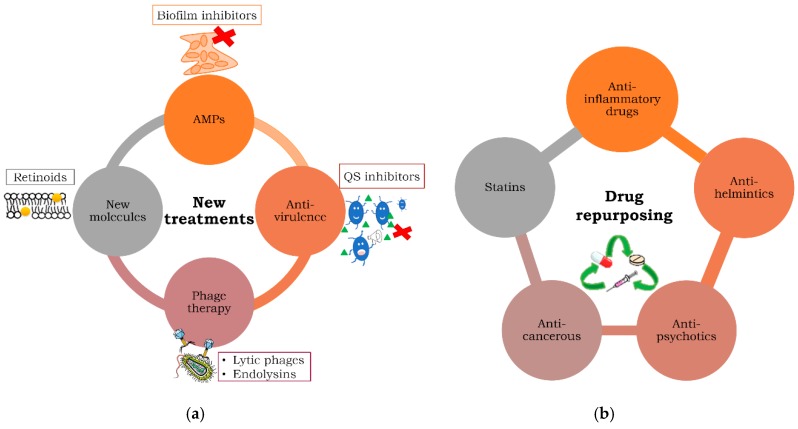
Anti-infectious strategies investigated with the aim of combating resistant and persistent bacteria. (**a**) Clockwise from the top: AMPs: antimicrobial peptides, including biofilm inhibitors; QS: quorum sensing. The green triangles illustrate the acyl-homoserin lactones that bacteria secrete as QS modulators. Phage therapy includes all the clinical trials in which a combination of lytic phages or parts of these (endolysins) are used against bacterial infections with therapeutic aims. Finally, new molecules encompass all synthetic compounds displaying antibacterial activity, such as retinoids (indicated by yellow circles inserted in the membrane). (**b**) Repurposing encompasses all the FDA (Food and Drug Administration)-approved drugs with potentially antibacterial effects.

**Figure 2 antibiotics-09-00065-f002:**
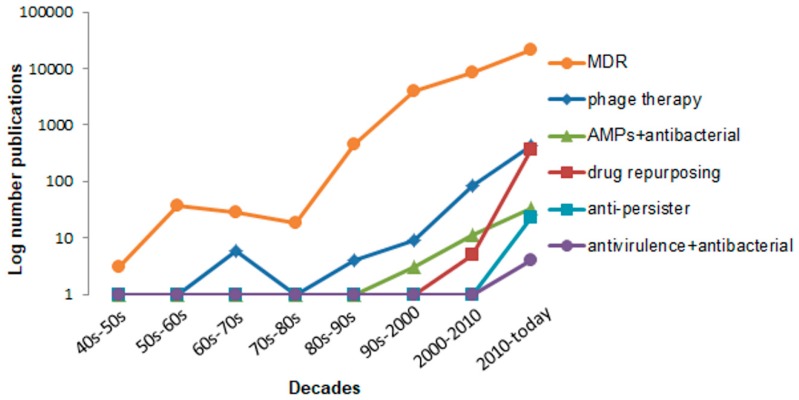
Number of annual publications available on PubMed search engine for the keywords “MDR OR multidrug resistance”, “phage therapy”, “antimicrobial peptides AND antibacterial”, “drug repurposing”, “anti-persister OR anti-persistence OR resuscitating drugs”, “anti-virulence AND antibacterial”. The number of publications is expressed on a logarithmic scale.

**Table 1 antibiotics-09-00065-t001:** Anti-multidrug-resistance (MDR) molecules that show activity against multidrug-resistant bacteria.

Name of Drug	Type of Drug	Active against	Mechanism of Action	Reference
P5 and P9	AMPs	MRSA	Inhibition of biofilm, disruption of membrane integrity, downregulation of virulence genes	[32]
Cationic peptides	Inhibition of TCS *vanRS*	VRE	Restoration of vancomycin activity	[33]
Inhibitors of AcrAB–TolC	Inhibition of AcrAB–TolC	MDR *Escherichia coli*	Restoration of antibiotic activity, induction of spontaneous mutations, inhibition of DNA mismatch repair system	[37]
Cocktail of 9 lytic phages	Lytic phages	MDR *Acinetobacter baumannii*	Lysis of MDR *A. baumannii* cells	[43]
Ab105-2phiΔCI	Engineered phage	MDR *A. baumannii*	Lysis of MDR *A. baumannii* cells, synergy with carbapenems	[49]
LysAB2	Endolysin from a phage	MRSA	Bactericidal activity against MRSA, *A. baumannii*, and *E. coli*	[54]
Cbf-K_16_	AMP anti-inflammatory	Clarithromycin and amoxicillin-resistant *Helicobacter pylori* SS1	Inhibition of IL-8, downregulation of virulence and adhesion genes	[61]
5-Fluorouracil and 6-thioguanine	Anti-cancerous	*A. baumannii*	Pirimidin and purin analogues	[73]
Gallium	Anti-cancerous	MDR ESKAPE	Competition with Fe^+3^	[76]
Clomiphene	Anti-cancerous (SERM)	MRSA	Inhibition of UPPS, synergy with β-lactams	[79]
Simvastatin	Statin	MRSA skin infections	Anti-biofilm activity against staphylococcal biofilms in vivo	[84]

AMPs: antimicrobial peptides; MRSA: methicillin-resistant *S. aureus*; TCS: two component system; VRE: vancomycin-resistant *Enterococcus*; MDR: multi-drug resistant; ESKAPE: *E. faecium, S. aureus, K. pneumoniae, A. baumannii, P. aeruginosa, Enterobacter spp.* SERM: selective estrogen receptor modulator.

**Table 2 antibiotics-09-00065-t002:** Anti-persister molecules that show results against persistent bacteria.

Name of Drug	Type of Drug	Active against	Mechanism of Action	Reference (PMID)
P2* (Defensin-like peptide)	Permeabilizer	*Staphylococcus aureus*	Binding to DNA, biofilm inhibition	[31]
BF8*	QS inhibitor	*E. coli*	Disruption of *E. coli* biofilm, restoration of ofloxacin activity	[34]
ADEP4	ClpP activator	*S. aureus*	Degradation of hundreds of proteins	[35]
M64	QS inhibitor	*Pseudomonas aeruginosa*	Inhibition of PqsR, down-regulation of virulence genes	[36]
Cocktail of 6 lytic phages	Lytic phages	Recurrent *P. aeruginosa*	Lysis of *P. aeruginosa* bacterial cells	[44]
phi11 endolysin	Endolysin from a phage	*S. aureus*	Disruption of S*. aureus* biofilms and bactericidal activity	[50]
CHAP(K)	Endolysin from a phage	*S. aureus*	Removal of staphylococcal biofilms	[51]
PlyC	Endolysin from a phage	*Streptococcus spp.*	Removal of streptococcal biofilms	[52]
PlyE146	Endolysin from a phage	*E. coli*, *P. aeruginosa*, and *A. baumannii*	Disruption of *E. coli*, *P. aeruginosa* and *A. baumannii* biofilms	[53]
CD437* and CD1530*	Retinoids (analogues of vitamin A)	*S. aureus*	Membrane disruption	[55]
Cisplatin	Anti-cancerous	*E. coli* K-12, *S. aureus*, *P. aeruginosa*	Forms intra-strand DNA crosslinks	[78]
Mitomycin C	Anti-cancerous	Broad range of persisters	Forms inter-strand DNA crosslinks	[78]

***** Also efficient against MDR cells. QS: quorum sensing; ADEP4: acyldepsipeptide.

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
