# Peer review of "Strategies to Combat Multidrug-Resistant and Persistent Infectious Diseases"

_antibiotics, 2020, doi:10.3390/antibiotics9020065_

Round 1

Reviewer 1 Report

Dear author,
This manuscript touches a sensitive and interesting topic.
It is an elaborated and documented manuscript. I strongly believe that you should give us some more information about the inclusion and exclusion criteria that you applied on this manuscript.

Row no. 108 - please detail the abbreviation, when is used for the first time.

Row no. 395,570- please replace"..." with etc.

Please reformulate the section "Acknowledgments" and "Conflict of interest"

It was a pleasure to review such a well written manuscript!

Best regards!

Author Response

Dear author,
This manuscript touches a sensitive and interesting topic. It is an elaborated and documented manuscript. I strongly believe that you should give us some more information about the inclusion and exclusion criteria that you applied on this manuscript.

Thanks for these nice comments.

Row no. 108 - please detail the abbreviation, when is used for the first time.

I suppose you talk about the WT abbreviation, now detailed in row 77 (first time in the text)

Row no. 395,570- please replace"..." with etc.

Done

Please reformulate the section "Acknowledgments" and "Conflict of interest"

Done

It was a pleasure to review such a well written manuscript!

Best regards!

Reviewer 2 Report

Pacios et al, presented a review article on ‘Strategies to combat persistent infectious diseases’ for peer review.

The article focuses on multidrug resistant pathogens rather than the infectious diseases, so I would suggest to change the title as ‘Strategies to combat persistent multidrug resistant pathogens’.

The multidrug resistance is a major global issue, currently, we see a huge increase in drug resistance among the microorganisms and importantly, the resistant determinants are transmitting to other organisms through a horizontal gene transfer. The persistent and tolerant microorganisms are the major driving factors leading to the development of multi-drug resistant. The review identified some potential and viable strategies to control such persistent bacteria such as antimicrobial peptides, anti-virulence and phage therapy and new synthetic molecules and drug repurposing.

I would recommend to add exact mode of action (if available) of all the compounds referred in this manuscript.

Please see some comments, but I just highlighted a few.

Line 26- “we find” doesn’t make sense, perhaps say ‘ in this review we focus…..

Line 28- “we have” the authors are in fact reviewing, rephrase the words ‘we have’

Line 35- “MDR are pathogens can develop resistance to different antimicrobials”. Delete ‘are’

Line 36-“gene horizontal transfer” say horizontal gene transfer

Line 53- “MDR strains able to cause to” say MDR strain and able to cause

Line 55-57-“Resistance is the ability of bacteria to grow under antibiotic pressure and is due to the inherited mutations which affect the efflux pumps or to the drug target [8,9].”

This sentence is more generic/vague, I don’t understand

Line 71- “due to lack of cellular metabolism: protein synthesis is stopped”. Please delete: and say lack of cellular metabolism, effect or interference on DNA replication, transcription, translation and cell wall synthesis.

Line 92- rephrase

Line 128- “in order to some help to researchers and doctors”. Say in order to help some researchers….

Line 150- Line 81- “There is strong evidence”. There is a strong evidence

Line 150- All these fields are gaining currency in the scientific community from some decades ago, do we need this sentence

Line 194- “also this year” please indicate the year 2019 or ??

Line 217-specify the mode of action of antivirulence compounds

Line 227- “Concerning E. coli, it was the team of Frimodt-Møller [38]” authors mentioned “Frimodt-Møller” whereas the reference is El Meouche, I.; Dunlop, M.J. Heterogeneity in efflux pump expression, check is this the correct reference. It would be useful to add the name of efflux inhibitor you mention here.

Line 294, “one year ago Kim et al” specify the year

Line 375, “it was Anderson’s team…” why not say Anderson et al., year …

Line 447- “teichoic acid wall and peptidoglycan of S. aureus”. It should be teichoic acid and peptidoglycan of S. aureus cell wall

Line 457- “As statins are lipid-lowering molecules which display pleiotropic effects,

their potential antibacterial ability has been analysed:”

Spilt or rephrase the sentences and remove the colon at the end.

Line 486- “The first cause of deaths worldwide are microbial infections [86],” the statement is wrong, the firsts cause of death is heart diseases and cancer. The reference 86 is wrong

Line 603- acknowledge people or delete the acknowledgement template

Author Response

Pacios et al, presented a review article on ‘Strategies to combat persistent infectious diseases’ for peer review.

The article focuses on multidrug resistant pathogens rather than the infectious diseases, so I would suggest to change the title as ‘Strategies to combat persistent multidrug resistant pathogens’.

We have accepted your suggestion and entitle the review “Strategies to combat multidrug resistant and persistent infectious diseases’

The multidrug resistance is a major global issue, currently, we see a huge increase in drug resistance among the microorganisms and importantly, the resistant determinants are transmitting to other organisms through a horizontal gene transfer. The persistent and tolerant microorganisms are the major driving factors leading to the development of multi-drug resistant. The review identified some potential and viable strategies to control such persistent bacteria such as antimicrobial peptides, anti-virulence and phage therapy and new synthetic molecules and drug repurposing.

I would recommend to add exact mode of action (if available) of all the compounds referred in this manuscript.

We have summarized what is new about the modes of action of just some of the mentioned drugs (4th column in Table 1 and Table 2) because, unluckily, not all the exposed drugs have a well-known mode of action (line 500).

Please see some comments, but I just highlighted a few.

Line 26- “we find” doesn’t make sense, perhaps say ‘ in this review we focus…..

Done

Line 28- “we have” the authors are in fact reviewing, rephrase the words ‘we have’

Done

Line 35- “MDR are pathogens can develop resistance to different antimicrobials”. Delete ‘are’

Done

Line 36-“gene horizontal transfer” say horizontal gene transfer

Done

Line 53- “MDR strains able to cause to” say MDR strain and able to cause

Done

Line 55-57-“Resistance is the ability of bacteria to grow under antibiotic pressure and is due to the inherited mutations which affect the efflux pumps or to the drug target [8,9].”

This sentence is more generic/vague, I don’t understand

Now we have completed the definition of resistance, based on “Brauner, A.; Fridman, O.; Gefen, O.; Balaban, N.Q. Distinguishing between resistance, tolerance and persistence to antibiotic treatment. Nat Rev Microbiol 2016, 14, 320-330, doi:10.1038/nrmicro.2016.34”, hoping it is better now.

Line 71- “due to lack of cellular metabolism: protein synthesis is stopped”. Please delete: and say lack of cellular metabolism, effect or interference on DNA replication, transcription, translation and cell wall synthesis.

Done

Line 92- rephrase

Done, hoping it is clearer.

Line 128- “in order to some help to researchers and doctors”. Say in order to help some researchers….

Done

Line 150- Line 81- “There is strong evidence”. There is a strong evidence

Done

Line 150- All these fields are gaining currency in the scientific community from some decades ago, do we need this sentence

Sentence removed

Line 194- “also this year” please indicate the year 2019 or ??

Done, it was in 2019

Line 217-specify the mode of action of antivirulence compounds

Ok, it was added in the lines 207-208 “Antivirulence compounds are those inhibiting bacterial virulence factors, thus preventing the development of infection without affecting bacterial growth”

Line 227- “Concerning E. coli, it was the team of Frimodt-Møller [38]” authors mentioned “Frimodt-Møller” whereas the reference is El Meouche, I.; Dunlop, M.J. Heterogeneity in efflux pump expression, check is this the correct reference. It would be useful to add the name of efflux inhibitor you mention here.

Absolutely right, the correct reference is El Meouche. Changed

Line 294, “one year ago Kim et al” specify the year

Done

Line 375, “it was Anderson’s team…” why not say Anderson et al., year …

Done

Line 447- “teichoic acid wall and peptidoglycan of S. aureus”. It should be teichoic acid and peptidoglycan of S. aureus cell Wall

Changed

Line 457- “As statins are lipid-lowering molecules which display pleiotropic effects,

their potential antibacterial ability has been analysed:”

Spilt or rephrase the sentences and remove the colon at the end.

Sentences have been split.

Line 486- “The first cause of deaths worldwide are microbial infections [86],” the statement is wrong, the firsts cause of death is heart diseases and cancer. The reference 86 is wrong

Reference removed.

Line 603- acknowledge people or delete the acknowledgement template

Done

Reviewer 3 Report

Antimicrobial-resistant infections represent a large burden on our global healthcare system. While it appears that this review would like to highlight treatments that impact bacterial persisters rather than discuss the broad topic of overcoming resistance, the link of the treatments presented specifically to addressing persister cells is lacking. 

When writing scientific reviews, it is imperative to use your knowledge of the field to summarize important findings and concepts in your own words. 

Extensive English editing is required to increase the clarity of this work. English errors have made some statements scientifically incorrect.  The short paragraph structure makes the work choppy. Casual language should be replaced with a scholarly tone.  The word exposition is used in numerous places where I believe that you mean exposure MDR is an adjective that must be used to modify a noun such as bacteria.  Sentences in lines 45 and 46 contradict each other The concept of antibiotic resistance is grossly oversimplified  Random bolding of words is distracting Figures are blurry and need to be saved/inserted as a higher resolution format Figure 1  - Drug repurposing with arrows suggests that a cycle is being used which is not the case. The drug classes listed are not related. Figure 2 - Without further filtering, several of your search queries will yield results that do not relate to treatment of infectious diseases and/or antimicrobial-resistant pathogens Lines 276-280 are grey instead of black text Line 395 = Unfinished sentence  Introductory sentence to Table 1 & 2 states that the table will summarize the treatments presented in text however not all in-text treatments can be found in the table.  Line 519 does not correctly summarize the paragraph.  References missing in the discussion section Bacterial names should be italicized and capitalized appropriately  Email addresses should only be included for the corresponding author. References need to be fixed. Several instances of duplicate brackets next to each other.  Acknowledgment and Conflict of Interest Statements are incomplete. 

Author Response

Antimicrobial-resistant infections represent a large burden on our global healthcare system. While it appears that this review would like to highlight treatments that impact bacterial persisters rather than discuss the broad topic of overcoming resistance, the link of the treatments presented specifically to addressing persister cells is lacking. 

We would like this a lot, but the majority of the works in literature focus on the analysis of anti-infectious treatments against MDR, but not specifically anti-persister treatments. In this review we have analyzed the anti-persister treatments until now (focusing on the molecular mechanisms described in relation to bacterial tolerance and persistence) against persistent bacteria. You can see this in the Table 2 as well as developed in the text.

Moreover, we did a new innovative review to difference our work from other paper published, such as “Fighting bacterial persistence: Current and emerging anti-persister strategies and therapeutics”, by Defraine et al. (2015), where authors exclusively developed "anti-persister treatments".

Our conclusion was that a possible via of study to fight persistent infectious bacteria would reside in analysing the coordination of several networks associated with molecular mechanisms of bacterial tolerance or persistence. The combination of new anti-infectious treatments, as well as drug repurposing alone or in association with antimicrobials, could be an efficient way to counter multidrug resistant (MDR) and persistent infectious bacteria.

When writing scientific reviews, it is imperative to use your knowledge of the field to summarize important findings and concepts in your own words. 

Absolutely, we have intended to do this and examples of that are lines 63-69, lines 118-121, 124-136, 170-175, 315-324, and mainly the discussion (even if here we have referenced many reviews in order to reinforce our own statements and analysis).

Extensive English editing is required to increase the clarity of this work. English errors have made some statements scientifically incorrect.

Ok, it has been revised

The short paragraph structure makes the work choppy.

We consider useful to organize our information in shorts paragraph where each one presents one single idea or drug, in order to facilitate the reading and the comprehension of this review.

Casual language should be replaced with a scholarly tone. 

Done (line 251)

The word exposition is used in numerous places where I believe that you mean exposure

Done

MDR is an adjective that must be used to modify a noun such as bacteria. 

Done (line 41)

Sentences in lines 45 and 46 contradict each other

Completely, one of them has been removed.

The concept of antibiotic resistance is grossly oversimplified 

As answered for Reviewer 2, we have now completed the definition of resistance, based on “Brauner, A.; Fridman, O.; Gefen, O.; Balaban, N.Q. Distinguishing between resistance, tolerance and persistence to antibiotic treatment. Nat Rev Microbiol 2016, 14, 320-330, doi:10.1038/nrmicro.2016.34”, hoping it is better now.

Random bolding of words is distracting

Bolding removed

Figures are blurry and need to be saved/inserted as a higher resolution format

Resolution of the figures has been enhanced

Figure 1  - Drug repurposing with arrows suggests that a cycle is being used which is not the case. The drug classes listed are not related.

Cycle format has been removed

Figure 2 - Without further filtering, several of your search queries will yield results that do not relate to treatment of infectious diseases and/or antimicrobial-resistant pathogens

The point of this figure is to illustrate the increasing interest on these topics (MDR pathogens, persisters, drug-repurposing) by researchers and scientists, from the 40s till nowadays, not just focusing on treatments (this is why we included the figure in the Introduction, to introduce the importance of the topic reviewed here)

Lines 276-280 are grey instead of black text

Done

Line 395 = Unfinished sentence 

We ended the list of virulence genes inhibited by niclosamide with the word “etc” instead of “…”, according to Reviewer 1

Introductory sentence to Table 1 & 2 states that the table will summarize the treatments presented in text however not all in-text treatments can be found in the table. 

True, we have now included all the in-text treatments that refer to persister or MDR cells.

Line 519 does not correctly summarize the paragraph. 

The paragraph has been slightly modified.

References missing in the discussion section

Added, lines 508 and 560

Bacterial names should be italicized and capitalized appropriately 

Done

Email addresses should only be included for the corresponding author.

We have followed the author instructions from this journal using the template.

References need to be fixed.

Done

Several instances of duplicate brackets next to each other. 

Several of these brackets have been removed and the clarifications/explanations have been flanked by commas (line 132, lines 174-176, lines 256-257, lines 316-317, lines 326-327, line 545, line 564)

Acknowledgment and Conflict of Interest Statements are incomplete.  

Done

Round 2

Reviewer 2 Report

Thanks for the corrections, the revision version looks nice.